# Smartphone Apps Targeting Hazardous Drinking Patterns among University Students Show Differential Subgroup Effects over 20 Weeks: Results from a Randomized, Controlled Trial

**DOI:** 10.3390/jcm8111807

**Published:** 2019-10-28

**Authors:** Anne H. Berman, Claes Andersson, Mikael Gajecki, Ingvar Rosendahl, Kristina Sinadinovic, Matthijs Blankers

**Affiliations:** 1Center for Psychiatry Research, Department of Clinical Neuroscience, Karolinska Institutet, & Stockholm Region Healthcare Services, Norra Stationsgatan 69, SE-11364 Stockholm, Sweden; mikael.gajecki@ki.se (M.G.); ingvar.rosendahl@ki.se (I.R.); kristina.sinadinovic@ki.se (K.S.); 2Stockholm Center for Dependency Disorders, Stockholm Region Healthcare Services, SE-112 81 Stockholm, Sweden; 3Department of Criminology, Malmö University, SE-205 06 Malmö, Sweden; claes.andersson@mau.se; 4Trimbos Institute—The Netherlands Institute of Mental Health and Addiction, 3500 AS Utrecht, The Netherlands; mblankers@trimbos.nl; 5Arkin Mental Health Care, 1033 NN Amsterdam, The Netherlands; 6Department of Psychiatry, Amsterdam UMC, Location AMC, University of Amsterdam, 1105 AZ Amsterdam, The Netherlands

**Keywords:** hazardous alcohol use, university students, smartphone apps, m-health, brief intervention

## Abstract

Overconsumption of alcohol, from hazardous to excessive, heavy, and harmful levels, is common among university students. Consenting Swedish students were assigned to one of two smartphone apps offering feedback on estimated blood alcohol concentration (eBAC; Promillekoll/PartyPlanner) or assessment only (*n* = 2166; 1:1:1 ratio). App participants with excessive drinking according to public health criteria (>9/>14 drinks/week for women/men, respectively) at a 7 week follow-up were additionally assigned to the skills-based TeleCoach app or waitlist (*n* = 186; 1:1 ratio). All participants were followed at 14 and 20 weeks. At 7 weeks, Promillekoll users showed higher risk of excessive drinking (odds ratio (OR) = 1.83; *p* ≤ 0.01; *n* = 1558). Students in eBAC app groups with only hazardous use showed fewer binge drinking occasions at 14 weeks and lower eBAC levels up to 20 weeks compared to controls (*n* = 1157). Also, more highly motivated participants at baseline in both eBAC app groups drank less compared to controls at 7 and 20 weeks. Hidden Markov model analysis revealed a frequent-heavy drinking group (*n* = 146; 4.6 days/week, SD = 1.4), where those with access to TeleCoach had fewer drinking days compared to assessment-only controls (*p* < 0.001). eBAC apps showed positive effects up to 20 weeks, particularly for motivated students, and a skills-based app can reduce consumption for those with frequent-heavy drinking patterns.

## 1. Introduction

Evidence has accumulated showing that overconsumption of alcohol, particularly heavy drinking over time, is associated with a high burden of disease with a range of negative personal, social, and economic effects, including higher risk of mental health conditions and high comorbidity of severe mental health disorders [1,2]. Although drinking in the hazardous range can be reduced on a population level through taxation measures [3], this type of drinking, including binge drinking, can also be effectively addressed on an individual level through screening and brief intervention measures [4,5]. This type of behavior is most frequent among adolescents and among young adults (18–29) [6], who are in a stage of “emerging adulthood” characterized by challenges of identity exploration, and management of emotional and circumstantial instability [7]. Although severity of alcohol consumption is often mitigated with entry into the job market and establishing a family, continued alcohol use is common and carries significant risk [4]. Treatment seeking-behavior is generally very low for alcohol-related and mental health problems among adults [8], and if intervention is possible at an early stage, it can prevent later development of more severe problems.

About half of young adults aged 18–25 attend college or university [9], a time when alcohol consumption and alcohol-related risks peak [10]. University students, who even at hazardous drinking levels tend to binge drink frequently [10], are thus an important target population for interventions, both within the student healthcare setting, as well as in ordinary clinical settings such as primary and emergency room care. Although effective treatment methods for individuals and groups are well-documented [11], very few students seek such help in any setting [12]. Indeed, students engaging in binge drinking at least once a month have indicated preferences for computerized interventions [13]. Very brief digital interventions, where students are given personalized feedback on their drinking levels in comparison to peer norms, have shown promise in a non-app format [14]. Also, digital interventions for alcohol problems, primarily internet-based, have been found to yield small but meaningful effects for adults, including college students [15,16]. When such digital interventions are evaluated in comparison with controls, they yield effect sizes similar to face-to-face counseling. However, in direct comparisons between digital and face-to-face interventions, the latter have emerged as more effective among student-aged populations [17,18]. Nonetheless, digital interventions hold the promise of much broader dissemination than face-to-face interventions and even small effect sizes could have considerable impact.

Commercially available native alcohol apps are growing exponentially in number, but the evidence for their effectiveness in reducing problematic alcohol use is limited, both in terms of few evaluations as well as negative or null effects in several trials [5]. A systematic review of mobile interventions based on text messages, automated telephony programs (IVR), and smartphone apps, all targeting hazardous alcohol use in university students, found only two studies with positive effects [19], one using IVR [20], and the second reporting positive results for a smartphone app based only on secondary outcomes [21]. Whether apps can contribute to reducing hazardous or even harmful drinking is not yet clear, and research to date is too sparse to draw any firm conclusions as to whether apps hold promise or not, suggesting that conducting further research is motivated. Indeed, different apps might have differential effects for different groups of students, possibly classified according to drinking patterns. Prior research has identified distinct drinking patterns among university students [22,23], and a 2007 review suggested that high-risk student groups might need specially targeted interventions [24]. Also, a later meta-analysis found that computer-delivered interventions could be more efficacious in the short term for heavier drinkers than for the student population at large, perhaps because heavier drinkers are not likely to recognize the severity of their drinking but respond with higher awareness of risk than those with lower risk levels, to the personalized feedback they access via computerized interventions [18,19,20,21,22,23,24,25,26,27,28]. 

Our research group has been studying smartphone apps in two consecutive studies including university students with hazardous drinking (see Figure 1). In Study 1 [25], participants who were hazardous drinkers, including frequent binge drinking, were randomly allocated to a native app or a web-based app, both offering feedback on real-time estimated blood alcohol concentration (eBAC) but where the web-based app also included the extra feature to plan and review drinking sessions; the apps were compared to an assessment only control group. At a 7 week follow-up, native app users had increased drinking frequency in comparison to the control group, whereas web-app users showed no differences in comparison to controls. It was also found that although all study participants were identified at baseline as hazardous drinkers with binge drinking once a week on average—at risk of developing alcohol-related problems—one-third were hazardous drinkers who also consistently drank excessively, that is, over recommended national norms. 

Study 2, for which the study protocol was published [26] therefore included two consecutive randomized controlled trials with a prolonged follow-up period up to 20 weeks. Following the results from our first study [25], the native and web-based apps were modified on the basis of user feedback, and a skills-training app was developed for hazardous drinkers who were identified at the 7 week follow-up as also characterized by consistent excessive drinking. For Study 2, only results for the latter, participants identified with excessive drinking at the 7 week follow-up, have been published [27]. The results showed that access to a skills-based app was associated with a significantly lower proportion of participants with excessive drinking at 14 and 20 week follow-ups compared to excessive drinkers in the waitlist group, with an overall odds ratio (OR) for no excessive drinking of 1.95 for the skills-based TeleCoach app group and an OR of 1.51 for the waitlist group (who gained access to the app at 14 week follow-up), where the reference group consisted of assessment-only controls [27]. 

The first aim of the present article was to explore whether revisions made to the native and web-based apps used in Study 1 would lead to reduced drinking frequencies at the 7 week follow-up for participants using either of the two apps in comparison to assessment-only controls. The second aim was to explore the native and web-based app effects, for up to 20 weeks, on alcohol consumption in hazardous drinkers with regular binge drinking but without excessive drinking, in comparison to assessment-only controls. The third and final aim was to draw overall conclusions from Study 2 by examining the interrelationship between individual drinking patterns and intervention-related behavior change in subgroups of participants, through hidden Markov modeling (HMM), a technique used to detect hidden trends in repeated measurements, applied occasionally in addiction research (e.g., [28]). 

The present article thus reports the parts of Study 2 that have not been previously published. Three analyses are presented. The first analysis, completely new, builds on data for the entire cohort, from baseline to the 7 week follow-up, a replication of Study 1 (Figure 1b) [25]. The second analysis, also completely new, includes data for hazardous—but not excessive—from 7 to 20 week follow-ups (Figure 1d). The third analysis using HMM builds on data for the entire cohort, from baseline to the 20 week follow-up (Figure 1b–d); this analysis is thus the only one in this study that includes the excessive drinkers who also participated in the randomized trial nested within Study 2 [27]. The HMM analysis contributed new findings on two partly overlapping groups of university students that both risk developing alcohol-related problems: hazardous drinkers characterized by occasional binge drinking, and hazardous drinkers with excessive drinking, who may differ in drinking patterns from hazardous drinkers without excessive drinking. The HMM analysis was intended to contribute to disentanglement of these two overlapping groups, who may differ from one another in more aspects than excessive drinking alone. 

## 2. Experimental Section

This was a three-armed trial evaluating the effectiveness of giving university students with hazardous alcohol use access to one of two smartphone apps offering feedback on real-time estimated blood alcohol concentration (eBAC) levels, with 7, 14, and 20 week follow-ups. Both intervention groups were compared to assessment-only controls. Participants who at the 7 week follow-up reported weekly alcohol consumption at levels above the national recommendations, that is, excessive drinking levels, were offered participation in an add-on randomized trial (see Figure 1c) where they were randomly assigned to a skills-training app or to a wait-list control group that received access to the app 7 weeks after recruitment. In this nested trial, participants were compared to an additional control group, comprised of participants with excessive drinking at 7 week follow-up who had been assigned to the assessment-only control group in Study 2 at recruitment [27]. 

Hazardous drinking was operationalized as having a baseline score on the Alcohol Use Disorders Identification Test (AUDIT; [29]) of at least 6 (women) or 8 (men) points [30]. Excessive drinking levels were operationalized as drinking in excess of Swedish national recommendations of ≤9 drinks/week for women or ≤14 drinks/week for men [31]. A CONSORT diagram over the trial is shown in Figure 2; participant flow for the add-on trial is reported elsewhere [27]. The study protocol was approved by the Stockholm Regional Ethics Vetting Board on 19 March 2014 (Ref. nr. 2014/278-31/2), and complied with the World Medical Association Declaration of Helsinki. Trial registration: NCT02064998, clinicaltrials.gov.

### 2.1. Procedure

Detailed information on procedure is available in the research protocol [26] and is briefly summarized here. University students from six universities or campuses in the capital area of Sweden were invited to participate via email and Facebook pages. Potential participants were informed that completing baseline and follow-up questionnaires in the study would entail participation in one trial and include them in a lottery with three iPad devices as prizes. The rationale behind the lottery was that it is one form of compensation commonly used as an incentive to increase response rates. In this study, all participants had an equal opportunity to win the lottery if they completed all follow-ups. An invitation to participate in the add-on randomized trial for those drinking excessively at 7 weeks [27] was sent directly after the first follow-up. Interested students received more information on the add-on study and the Swedish Personal Data Act via a web page, where they could indicate their consent to participate. 

Those who provided their informed consent also entered their mobile phone number, information on gender, age, and weight, and responded to the Daily Drinking Questionnaire (DDQ) [32], the AUDIT [29], and the Readiness Ruler, a rating of their motivation to change drinking levels, on a scale from 1 (no motivation) to 10 (highest motivation) [33]. Participants fulfilled inclusion criteria if they had at least hazardous drinking levels according to the AUDIT and possessed an iOS or Android smartphone device; these were randomized at a 1:1:1 ratio to one of the two app conditions or to an assessment-only control group, using the randomization function in IBM SPSS Statistics for MacOS X, Version 22 (IBM Corp., Armonk, NY, USA). All participants were informed that some students would receive an e-mail containing a link to a smartphone app, and that all would be asked to fill in follow-up questionnaires at approximately 7, 14, and 20 weeks after registration. No personalized feedback on baseline consumption levels was given to any of the participants. At 7 and 14 week follow-ups, participants responded to the DDQ. At 20 week follow-up they responded to the DDQ, the AUDIT, and to questions about user experience of interacting with the apps, and about access to other types of treatment during the study period, such as additional sources of information, medication, speaking to counselors, or using apps that were not part of the study. Primary outcomes were quantity, frequency, and number of binge drinking occasions, as well as mean and peak eBAC, and proportion of individuals drinking over the recommended levels of 9 and 14 drinks per week, for women and men, respectively. eBAC calculations, considered accurate for both low and high levels of intoxication in student populations [34], were made according to the Widmark formula, using the procedure described in Study 1 [25]. 

Participants in either of the smartphone intervention groups who reported excessive weekly alcohol consumption at the 7 week follow-up (*n* = 257) were informed about their excessive drinking levels and invited to participate in the add-on trial. The 186 participants who gave their consent for this trial were randomized into an intervention or a wait-list group at a 1:1 ratio. Assessment-only controls were extracted from participants in the main trial who had had excessive drinking levels at recruitment [27]. Excessive drinkers in the add-on trial were followed up at 14 and 20 weeks, in parallel with follow-ups for the remaining non-excessive drinkers. Waitlist participants were given access to the skills-based app at the 14 week follow-up [27].

### 2.2. Interventions

All app interventions were web-based and are described in detail in the research protocol [26]; a brief summary is given here. In terms of behavior change components, all three apps offer information on protective behavioral strategies (PBS) [35]. The Promillekoll app, a native app produced by the Swedish government-owned alcohol retail monopoly, *Systembolaget*, and the PartyPlanner app, a web-based app developed by the research group, both offer real-time feedback on eBAC levels via user-entered information on drinks per hour in addition to PBS. On top of this, the PartyPlanner app adds the functionality of simulating or planning a drinking event beforehand and then comparing the simulation to the real-time event afterwards. The TeleCoach app, a web-based app also developed by the research group, offers registration of alcohol intake over time as well as multiple specific skills training in mini-modules entitled “Saying no to alcohol” and “Feeling better without alcohol”. 

### 2.3. Statistics

Descriptive statistics on baseline characteristics are presented, with analyses of variance used to indicate any differences between groups. First, to analyze whether the revised versions of the two apps used in the preceding study yielded the same results in the second study, a linear mixed models analysis was performed, using the entire sample (*n* = 2166) for analysis of effects at the 7 week follow-up. Second, to analyze the effects of the native and web-based apps in non-excessive drinkers (*n* = 1157), defined as those drinking 9 or fewer standard drinks per week (women) or 14 or fewer drinks per week (men) at the 7 week follow-up, we conducted a linear mixed model analysis of effects over four timepoints (baseline, 7, 14, and 20 week follow-up). Additionally, a three-level model was applied, with the six different universities constituting the clusters at the highest level (level three). 

The 1157 students constituted the second level. Level one consisted of repeated measures over time, with outcomes expressed as five drinking parameters: *quantity* (standard glasses/week), *frequency* (drinking occasions/week), *binge occasions per week*, *eBAC per week*, and *peak eBAC per month*. A separate model was run for each of the five drinking parameters. Motivation to change drinking habits was added as a covariate parameter. The contribution to explaining the variance in the data by adding a third level, as compared to a two-level model, was tested with a likelihood-ratio test; the same test was used to compare a random-slope model to a random-intercept model in terms of data fit. The random coefficient was included at level two since the randomization to the apps was applied at the individual (student) level. Regarding justification of the three-level model with a random intercept and a random coefficient, it should be noted that the likelihood-ratio test comparing a random-coefficient model to a random intercept model yielded a highly significant chi-square value of 75.96 (degrees of freedom (DF) = 1, *p* = 0.000). The intra-class correlation value on the third level in an empty model was as low as 0.014. However, a comparison of the three-level model to the two-level model with a likelihood-ratio test yielded a chi-square value of 9.50, which was also highly significant (DF = 1, *p* = 0.000).

Third, we conducted an HMM analysis [36] on the entire sample (*n* = 2166) to identify hidden (latent) patterns in student drinking behavior over 20 weeks, and to evaluate whether these patterns might be associated with stable groups (classes) of students as well as with app assignment. For this analysis, the day-by-day weekly drinking records for all participants were used, for one specific week at each of the four timepoints. Hence, in total, 7 × 4 = 28 daily alcohol use quantity observations (1 per day) were available for each participant. The HMM is essentially a sequence model. Using HMM, a number of hidden states or classes were identified in the data on the basis of the characteristics of each student’s sequence of 28 drinking records. Each of the students was initially associated with one of the identified states. On the first day for which drinking data was collected for each student (for example, Monday), each student was associated with a certain state. However, students were able to transition from state to state. To model this process, a transition matrix was inferred from the data, which described the probability that a student moved from one state to the next state, for any given day on which drinking data were collected. The HMM is a probabilistic model in the sense that it computes the best-fitting state and the probabilities in the transition matrix on the basis of a probability distribution [37,38]. For this analysis, we used the R package depmixS4 to perform the HMM analysis. The optimal number of classes in the data was obtained by minimizing the Bayesian information criterion (BIC) statistic. 

The response rates ranged from a low of 55% for the PartyPlanner group at the 14 week follow-up to a high of 83% at the 7 week follow-up for the assessment-only control group, rates that are not atypical for app studies among university students where no non-automated human contact occurs [25,27]. Missing observations were not imputed; both the mixed models and HMM implementations are capable of handling missing data by making maximum likelihood (ML) estimations under the assumption of missingness at random (MAR). The MAR assumption is common in clinical epidemiological research, and considered to be tenable (e.g., [39]). 

## 3. Results

### 3.1. Baseline Characteristics 

A total of 4615 students provided their informed consent. Participant characteristics for those who fulfilled criteria for inclusion at baseline (*n* = 2166) are shown in Table 1. Women predominated in the sample (67.5%; *n* = 1462) and the mean age was just under 26 years (25.84, SD = 6.57). The mean AUDIT score was 10.98 (SD = 4.33), whereas the mean consumption of standard drinks per week was 9.4 (SD = 6.69). On average, participants reported 2.39 (SD = 1.23) drinking occasions per week, with the number of binge drinking occasions per week averaging 1.04 (SD = 0.91). The eBAC level per week was on average 0.018 (SD = 0.023), whereas the mean peak eBAC over the past month was 1.35 (SD = 0.89). With regard to the national recommended drinking levels, 31% of the participants drank excessively at baseline, that is, over the national recommended levels. The baseline level of motivation to change drinking habits was 4.07 (SD = 2.61). A priori drinking frequency was slightly lower in the PartyPlanner group compared to the control group (M = 2.31, SD = 1.21 vs. M = 2.48 SD = 1.24; *p* = 0.026); otherwise the groups did not differ in any of the characteristics.

### 3.2. 7 Week Follow-up

The Promillekoll group showed a higher relative risk over 7 weeks for excessive drinking in comparison to controls, controlled for the average motivation to change on a group level (OR 1.83; *p* ≤ 0.01; CI = 1.02; 2.99). In comparisons between the PartyPlanner group and controls, there were no significant differences (see Table 2).

### 3.3. 7, 14, and 20 Weeks for Non-Excessive Drinkers

Analysis of the non-excessive drinking group (*n* = 1157) at baseline, 7, 14, and 20 week follow-ups indicated that eBAC levels per week were lower for both the PartyPlanner and Promillekoll groups compared to controls at 7, 14, and 20 weeks. Binge occasions were lower for the Promillekoll group compared to controls at 7 and 20 weeks. At 14 weeks, binge occasions were lower for both groups compared to controls (see Table 3).

### 3.4. Motivation to Change

The three-level model showed that motivation at baseline had a reduction effect on drinking among the group of non-excessive drinkers (*n* = 1157). Motivation to change drinking habits at baseline was significantly associated with lower quantity of alcohol consumption per week for both app groups at 7 and 20 week follow-ups compared to controls (7 week follow-up: Promillekoll: z = −3.85; *p* = 0.000; contrast = −0.39; SE = 0.010; CI: −0.41; −0.37; PartyPlanner: z = −3.17; *p* = 0.002; contrast = −0.26; SE = 0.008; CI: −0.28; −0.24) and 20 week follow-up (Promillekoll: z=−2.05; *p* = 0.040; contrast = −0.37; SE = 0.018; CI: −0.41; −0.33; PartyPlanner: z = −5.87; *p* = 0.000; contrast = −0.06; SE = 0.009; CI: −0.08; −0.04). 

### 3.5. Hidden Markov Model Analysis

The HMM analysis of all participants over time (*n* = 2166) revealed the existence of eight ‘hidden’ states in the data according to the model with the lowest BIC value. Examination of the transition matrix indicated that seven of these states were associated with drinking patterns on one specific weekday (Monday–Sunday; see Appendix A). The eighth hidden state, termed *frequent-heavy* in line with earlier studies [23,40], revealed that 146 of the participants had a clearly distinctive drinking pattern that differed from all other students (see Table 4). Students in this very stable state (as inferred from the transition matrix in Appendix A) drank several days a week, with excessive drinking levels over the national recommendations. In the frequent-heavy state, 119 individuals were continual members, meaning that they drank several days a week as a consistent behavioral pattern over the 20 weeks of the study. In contrast, the remaining 27 were itinerant members of the frequent-heavy hidden state, meaning that they sometimes drank according to the frequent-heavy pattern, but at some point transitioned to one of the other hidden states. Table 4 shows baseline values for the continual and itinerant members of the frequent-heavy state, with comparisons between all frequent-heavy state members (*n* = 146) and the remaining sample (*n* = 2020). In summary, the frequent-heavy state differed from the rest of the cohort in that it included a higher proportion of men (52.1% vs. 31.1%) and older members (means of 29.5 vs. 25.6 years). In terms of alcohol consumption, the AUDIT score at baseline was higher (13.8 vs. 10.8), with higher quantity (16.4 drinks/week vs. 8.9) and frequency (4.6 occasions/week vs. 2.2). Also, the response rates of the frequent-heavy state members were somewhat higher than those of the remaining sample at the 7 week follow-up (79.5% vs. 71.4%), 14 week follow-up (76.0% vs. 64.6%), and at the 20 week follow-up (75.3% vs. 62.7%).

Associations between frequent-heavy drinkers and app assignment showed that frequent-heavy drinkers (either itinerant or continual) in the control group reduced their probability of drinking on any of the days per week to a lesser extent over time compared with those who had access to the Promillekoll app (OR = 1.03; SE = 0.01; CI = 1.01–1.05; *p* = 0.004). There were no differences between those who had access to the PartyPlanner app and those who had access to the Promillekoll app (OR = 1.01; SE = 0.01; CI = 0.99–1.04; *p* = 0.270) (see also Figure 3A and Appendix A). 

Frequent-heavy drinkers who were in the assessment-only control group had a less pronounced downward slope in terms of reduced drinking days compared to TeleCoach participants (OR = 1.06; SE = 0.02; CI = 1.03–1.09; *p* < 0.001). The difference in downward slope between waitlisted control participants and TeleCoach participants was not statistically significant (OR = 1.03; SE = 0.02; CI = 0.99–1.08; *p* = 0.144; see also Figure 3B and Appendix A). 

## 4. Discussion 

This study contributed findings on differential effects for apps targeting hazardous drinkers, with both non-excessive and excessive drinking. Overall, eBAC apps had a positive effect for non-excessive drinkers up to 20 weeks, and positive app effects were found for frequent-heavy drinkers, who benefited over time from access to the Promillekoll eBAC app, as well as from the TeleCoach skills-training app in comparison to assessment-only controls. However, access to the Promillekoll app was associated with a higher risk of excessive drinking compared to the control group over 7 weeks, in alignment with our prior finding of a short-term negative effect for Promillekoll app users [25]. Nonetheless, hazardous but non-excessive drinkers followed up to 20 weeks showed lower eBAC levels in both intervention groups, compared to controls, at all three follow-ups, with lower binge drinking for the Promillekoll group at all three follow-ups, as well as at the second follow-up for the PartyPlanner group. Secondary analyses showed that higher motivation at baseline to reduce drinking was associated with lower quantity of consumption over time for both the Promillekoll group and the PartyPlanner groups at 7 and 20 week follow-ups relative to assessment-only controls. Also, this study revealed the existence of a group of frequent-heavy drinkers who benefitted both from access to an eBAC app as well as to a skill-based app. This group included more males, older students, and daily drinkers compared to the rest of the participant cohort, consisting of both non-excessive and excessive drinkers. For frequent-heavy drinkers, access to the Promillekoll app appeared to be associated with lower quantities of drinking over time. Additionally, access to the TeleCoach app (available from the 7 week follow-up onwards) showed a robust association with fewer drinking days for the frequent-heavy drinkers.

Study strengths included a design following a stepped care approach, and identification of a hidden sub-group that differed in baseline characteristics and intervention response from other participants. Because the overwhelming majority of apps on the market have not been the subject of any research at all, this study makes a major contribution to the literature by beginning to disentangle relationships between time, severity of use, individual drinking patterns, and intervention content. In terms of limitations, the study was limited by reliance on self-report data, lack of information on actual app use, attrition, and a complex study design. Retention of participants in the study may have been mediated by motivation to reduce drinking, thus reducing external validity in relation to the university student population; however, motivation was analyzed as an independent factor influencing the study findings, thus mitigating the effect of any biased retention. Regarding self-report data, limitations were mitigated by use of questionnaires having been validated in Swedish and online contexts, as well as the fact that participants had no obvious incentive to underreport their drinking behaviors. Regarding information on app use, this has previously been assessed via self-report [25], but due to the use of multiple apps in this study, which participants did not recall the names of, the self-report data collected could not be used. Also, we were unable to allocate resources to make app-based user data directly available; this limitation has been remedied for coming studies. Regarding management of missing data, despite the fact that later records in longitudinal data with missing data are deleted, the maximum likelihood (ML) method borrows information from the values of the dependent variable in the earlier records to project what would happen later on during follow-up. At the same time, ML fully accounts for the uncertainty of this projection in the calculation of the standard errors and test statistics. In other words, groups with more missing data receive higher standard errors and become less likely to statistically be different from other groups with more complete data. 

This study had a complex design but contributed a framework for distinguishing between university students who are non-excessive and excessive drinkers, yielding findings suggesting that their need of support differs and that research in this area should attend to distinct sub-group needs. Specifically, our findings suggest that app-based interventions for university students could benefit from specific adaptations to student characteristics, particularly engagement, motivation, and problem severity. First, given the positive effects identified over a period of up to 20 weeks, in contrast to the negative and null findings over 7 weeks, apps targeting university students should include features designed to engage and maintain use over a longer period of time than a few weeks. A recent systematic review identified 19 randomized, controlled trials on apps targeting problematic alcohol use [41]; the authors’ qualitative synthesis described context, theoretical base, delivery mode, content, and implementation procedure as five core components that could potentially influence results. The success of text messaging and interactive voice response delivery modes suggest that push notifications might enhance app effects in the future [41]. Second, the importance of individual motivation to reduce drinking for positive effects, in combination with eBAC apps, suggests that apps should include a measure of motivation to change and offer feedback indicating that higher motivation is associated with actual positive change. Initial information on app use could include a warning that short-term use may be associated with negative or null results, and that longer-term use is recommended. Non-motivated users might benefit from motivational app interventions as a precursor to use of apps with the type of content we evaluated. Third, university students with frequent-heavy drinking patterns seem to gain more benefit from skills-based apps than from apps focused on eBAC levels, although a native eBAC-based app did contribute some benefit over time in terms of reduced drinking levels in this group. This is in concordance with current research on digital interventions for problematic alcohol use showing that more intense programs, including, for example, modules on refusal skills and management of negative emotions, tend to have slightly better effects on problematic use [15,16].

Future research should target university students as well as community and clinical samples of treatment-seeking adults. While our findings concern adult university students with baseline AUDIT scores in the hazardous zone, the results may also apply to adults with excessive drinking and higher AUDIT scores who are seeking help via internet interventions [16,42,43]. Future studies for both student and adult populations with hazardous and excessive drinking should apply innovative design to offer differential interventions for varying drinking patterns, paying particular attention to the needs of frequent-heavy excessive drinkers, whose alcohol consumption is likely to be associated with the development of alcohol use disorders and related comorbid physical and mental conditions. The role and benefit of using an eBAC-based app in combination with a skills-based app for reducing drinking levels could be more fully explored in future studies of frequent-heavy student drinkers, but also as a possible component in treatment of adult drinkers with more complex problematic alcohol use. For university students, a prior study has demonstrated positive effects of combining motivational counseling with feedback to reduce heavy drinking among students [44], suggesting a possible catalytic role for human guidance in combination with digital interventions for university students. Help-seeking adults may be more motivated to change their behavior than university students, yet guidance has shown a small but significant advantage in terms of reducing problematic alcohol consumption [16].

## 5. Conclusions

In summary, future research should begin to disentangle differential effects of eBAC and skills-based apps, more extended digital programs, and guidance versus no guidance, in relation to baseline characteristics including severity of alcohol use, drinking patterns, and motivation. The effects of combining apps with clinical face-to-face treatment in blended care is an emerging area of research that is in its infancy and holds significant promise in the area of addictions as well as mental health issues [45]. We look forward to a time when problematic alcohol use and other addictive behaviors will be met with evidence-based blended care in clinical contexts, minimizing the long delay often typical of addiction services, wherein patients access care after years of suffering for themselves and those around them. 

## Figures and Tables

**Figure 1 jcm-08-01807-f001:**
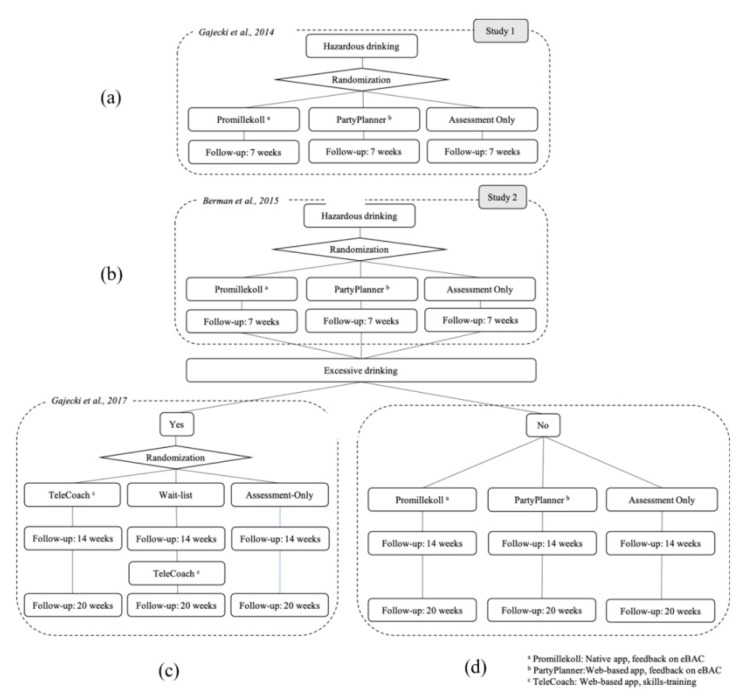
Study protocols and results from app Studies 1 and 2, reported as in panels a–d: (**a**) the research protocol and results for Study 1 were reported in Gajecki et al., 2014 [25]; (**b**) the research protocol for Study 2 was reported in Berman et al. 2015 [26]; (**c**) the results from the Study 2 add-on trial for excessive drinkers were reported in Gajecki et al. 2017 [27]; and (**d**) the results, reported in the current study, from (i) the entire cohort followed between baseline and a 7 week follow-up, (ii) hazardous but non-excessive drinkers followed between a 7 and 20 week follow-up, and (iii) the entire cohort, including participants with excessive drinking shown in panel (**c**), followed between baseline up to the 20 week follow-up.

**Figure 2 jcm-08-01807-f002:**
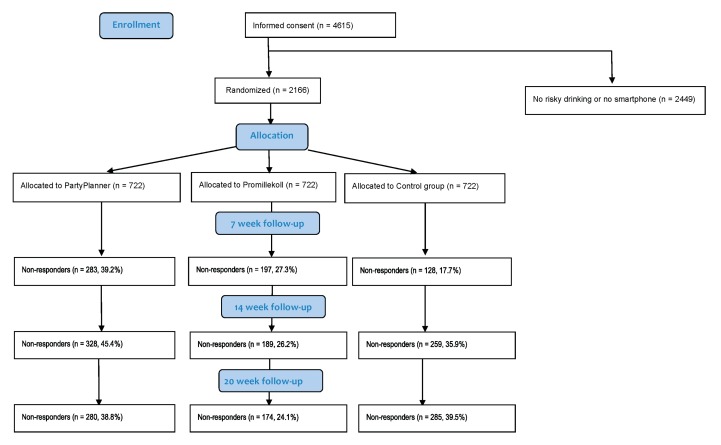
CONSORT diagram showing participant flow.

**Figure 3 jcm-08-01807-f003:**
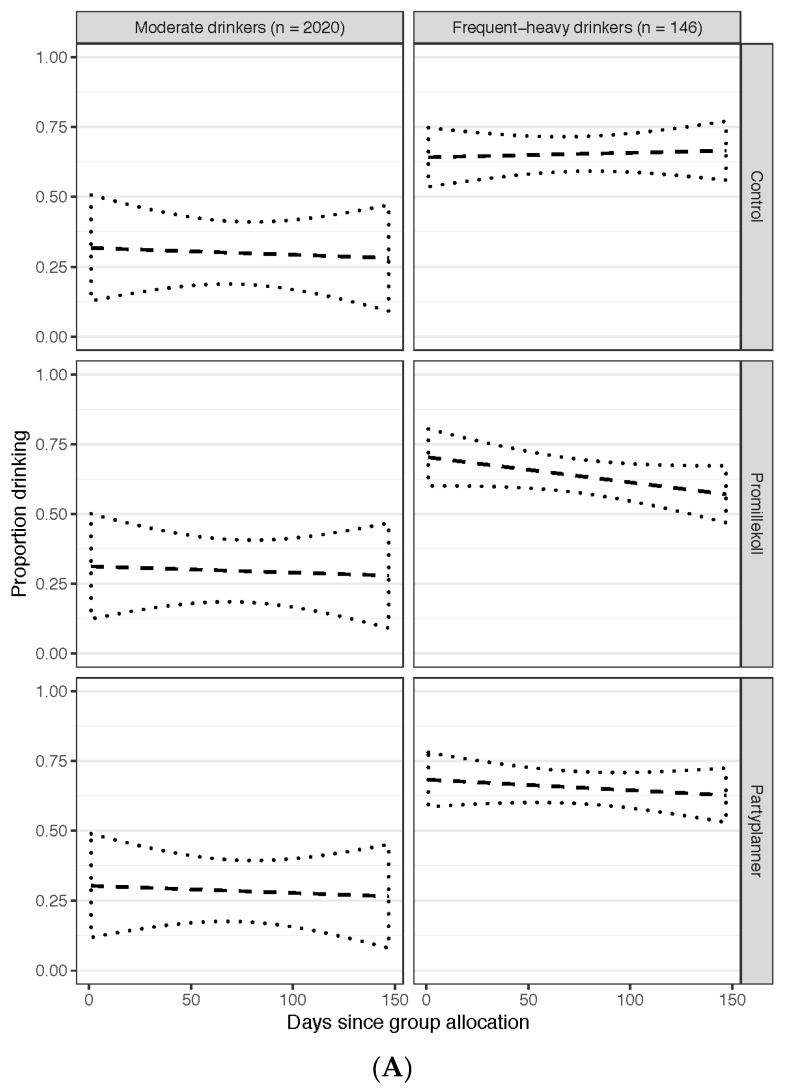
The proportion of students drinking alcohol (vertical axis) during any of the measurement weeks (horizontal axis; baseline, 7, 14, and 20 week follow-ups). (**A**) The left-hand column comprises three plots presenting data from moderate drinkers in the assessment-only control condition), Promillekoll eBAC app, and PartyPlanner eBAC app; the right-hand column presents the data from the frequent-heavy drinkers in the same study arms. (**B**) The left-hand column comprises three plots presenting data from moderate drinkers in the wait list, TeleCoach, and assessment-only conditions; the right-hand column presents the data from the frequent-heavy drinkers in the same study arms. A linear trend line (dashed) with 95% confidence interval of the trend line (dotted) has been included in each plot.

**Table 1 jcm-08-01807-t001:** Baseline characteristics of students with risky alcohol use in a randomized brief intervention smartphone app trial.

Characteristic	Total (*n* = 2166)	PartyPlanner (*n* = 722)	Promillekoll (*n* = 722)	Control (*n* = 722)	*p*-Values ^a^
Gender: M (%)/F (%)	704 (32.5)/1462 (67.5)	206 (31.3)/496 (68.7)	226 (31.2)/497 (68.8)	253 (35.0)/469 (65.0)	0.203
Age: mean (SD)	25.84 (6.57)	25.73 (6.38)	25.84 (6.36)	25.93 (6.94)	0.844
Measures of alcohol consumption: means (SD)					
Alcohol Use Disorders Identification Test (AUDIT) score (scale 0–40)	10.98 (4.33)	10.98 (4.40)	10.96 (4.21)	10.99 (4.38)	0.992
Quantity (standard glasses/week)	9.40 (6.69)	9.24 (6.57)	9.24 (6.44)	9.71 (7.05)	0.309
Frequency (drinking occasions/week)	2.39 (1.23)	2.31 (1.21)	2.37 (1.25)	2.48 (1.24)	0.026 ^b^
Binge occasions (no. per week)	1.04 (0.91)	1.03 (0.90)	1.04 (0.91)	1.04 (0.94)	0.985
Average estimated blood alcohol concentration (eBAC) ^c^ per week	0.018 (0.023)	0.019 (0.017)	0.018 (0.020)	0.017 (0.030)	0.487
Peak eBAC ^d^ within past month	1.35 (0.89)	1.40 (0.93)	1.33 (0.87)	1.32 (0.86)	0.197
Percent (%) over weekly recommended level	31.0	30.3	30.5	32.1	0.709
Motivation to change drinking behavior (scale 0–10)	4.07 (2.61)	4.06 (2.63)	4.17 (2.67)	3.97 (2.54)	0.358

^a^: *p*-values are based on ANOVA for age, AUDIT, quantity, frequency, binge occasions, average eBAC and peak eBAC; Pearson’s chi-square was used for gender and percent over weekly recommended level. ^b^: The source of the difference is between PartyPlanner and the control group. ^c^: Estimated average percentage blood alcohol count (BAC) per week. ^d^: Estimated average peak percentage BAC within the past month.

**Table 2 jcm-08-01807-t002:** Baseline and 7 week follow-up alcohol consumption outcomes, comparing intervention groups to controls (*n* = 1558).

Measures of Alcohol consumption in Means (SD); *N* = 2166	Control		PartyPlanner		PartyPlanner Compared to Control Linear Mixed Models ^c^	Promillekoll		Promillekoll Compared to Control Linear Mixed Models ^d^
	Baseline (*n* = 722)	Follow-up (*n* = 594)	Baseline (*n* = 722)	Follow-up (*n* = 439)	Time*Group *p*-Value	Baseline *n* = 722)	Follow-up (*n* = 525)	Time*Group *p*-Value
Quantity (standard glasses/week)	9.601 (6.669)	8.634 (6.313)	9.268 (6.277)	8.351 (5.838)	0.828	9.283 (6.149)	8.875 (6.559)	0.135
Frequency (drinking occasions/week)	2.472 (1.211)	2.390 (1.306)	2.310 (1.192)	2.238 (1.201)	0.799	2.367 (1.234)	2.349 (1.210)	0.101
Binge occasions (no. per week)	1.032 (0.919)	0.833 (0.841)	1.033 (0.881)	0.863 (0.880)	0.778	1.044 (0.891)	0.879 (0.884)	0.726
eBAC ^a^/week	0.019 (0.018)	0.017 (0.016)	0.019 (0.017)	0.017 (0.017)	0.765	0.019 (0.018)	0.018 (0.018)	0.208
Peak eBAC ^b^/month	1.331 (0.821)	1.238 (0.842)	1.393 (0.903)	1.266 (0.915)	0.419	1.331 (0.843)	1.270 (0.904)	0.694
Risk for excessive weekly drinking	0.322 (0.467)	0.244 (0.420)	0.302 (0.458)	0.253 (0.412)	Odds ratio (OR) 1.33 ^n.s.e^	0.305 (0.460)	0.284 (0.431)	OR 1.83 **

^a^: Estimated average percentage BAC for the week. ^b^: Estimated average peak percentage BAC during the past month. ^c^: Equivalent results to Study 1 [25]. ^d^: Non-equivalent results to Study 1 [25], with increased risk for excessive drinking rather than increased drinking frequency. ^e^. Non-significant (^n.s.^).** *p* ≤ 0.01.

**Table 3 jcm-08-01807-t003:** Baseline and 7, 14, and 20 week follow-ups for participants with hazardous but non-excessive drinking levels, defined as participants with a baseline AUDIT score of ≥6 W/≥8 M, and baseline weekly drinking levels below the definition of excessive drinking (>9 W/>14 M drinks per week).

Measures of Alcohol Consumption in Means (SD) *N* = 2166 ^a^	Control		PartyPlanner		PartyPlanner Compared to Control Linear Mixed Models	Promillekoll		Promillekoll Compared to Control Linear Mixed Models
	Baseline	Follow-up	Baseline	Follow-up	Time*Group *p*-Value	Baseline	Follow-up	Time*Group *p*-Value
**7 week follow-up for hazardous drinkers, *n* = 1157**	(*n* = 450)	(*n* = 450)	(*n* = 335)	(*n* = 335)		(*n* = 372)	(*n* = 372)	
Quantity (standard glasses/week)	7.540 (4.454)	5.790 (3.228)	7.081 (4.082)	5.380 (3.137	0.860	7.161 (4.184)	5.489 (2.961)	0.686
Frequency (drinking occasions/week)	2.227 (1.115)	2.013 (1.141)	2.002 (1.035)	1.827 (1.103)	0.064	2.093 (1.157)	1.937 (1.081)	0.085
Binge occasions (no. per week)	0.782 (0.755)	0.487 (0.562)	0.804 (0.707)	0.524 (0.595)	0.062	0.765 (0.718)	0.452 (0.562)	0.040 *
eBAC ^b^/week	0.014 (0.012)	0.011 (0.009)	0.014 (0.012)	0.010 (0.008)	0.001 ***	0.014 (0.010)	0.010 (0.008)	0.001 ***
Peak eBAC ^c^/month	Model did not converge							
Risk for excessive weekly drinking	Model did not converge							
**14 week follow-up for hazardous drinkers, *n* = 984**		(*n* = 401)		(*n* = 266)			(*n* = 317)	
Quantity (standard glasses/week)		6.672 (6.317)		5.950 (4.024)	0.128		6.014 (4.830)	0.072
Frequency (drinking occasions/week)		2.007 (1.320)		1.872 (1.142)	0.100		2.012 (1.324)	0.107
Binge occasions (no. per week)		0.630 (0.793)		0.599 (0.661)	0.046 *		0.543 (0.712)	0.022 *
eBAC ^b^/week		0.013 (0.015)		0.012 (0.012)	0.001 ***		0.011 (0.012)	0.001 ***
**20 week follow-up for hazardous drinkers, *n* = 975**		(*n* = 400)		(*n* = 262)			(*n* = 313)	
Quantity (standard glasses/week)		6.320 (4.306)		5.764 (3.992)	0.088		5.868 (3.885)	0.112
Frequency (drinking occasions/week)		1.922 (1.155)		1.805 (1.088)	0.093		1.847 (1.088)	0.093
Binge occasions (no. per week)		0.599 (0.762)		0.611 (0.676)	0.061		0.541 (0.674)	0.043 *
eBAC ^b^/week		0.012 (0.012)		0.011 (0.011)	0.001 ***		0.010 (0.010)	0.001 ***

^a^: Total number of participants in Study 2, with n shown for participants with hazardous but non-excessive drinking for each measurement occasion; ^b^: Estimated average percentage BAC for the week; ^c^: Estimated average peak percentage BAC during the past month.; * *p* ≤ 0.05; *** *p* ≤ 0.001.

**Table 4 jcm-08-01807-t004:** Baseline characteristics for participants in the frequent-heavy hidden state in comparison to other Study 2 participants with moderate drinking.

Characteristic	Continual Frequent-Heavy Drinkers (*n* = 119) ^a^	Itinerant Frequent-Heavy Drinkers (*n* = 27) ^a^	Total Frequent-Heavy Drinkers (*n* = 146)	Total Moderate ^b^ Drinkers (*n* = 2020)	t/Z	*p* ^c^
Gender: M (%)/W (%)	M = 63 (52.9%)/W = 56 (47.1%)	M = 13 (48.1%)/W = 14 (51.9%)	M = 76 (52.1%)/W = 70 (47.9%)	M = 628 (31.1%)/W = 1392 (68.9%)	Z = −5.2231	<0.0001
Age: mean (SD)	30.4 (10.2)	25.7 (4.1)	29.5 (9.5)	25.6 (6.2)	t = 4.9661 (df = 154.06)	<0.0001
Measures of alcohol consumption: means (SD)						
AUDIT score (scale 0–40)	14.1 (6.1)	12.4 (4.3)	13.8 (5.8)	10.8 (4.1)	t = 6.206 (df = 155.82)	<0.0001
Quantity (standard glasses/week)	17.3 (10.6)	12.4 (8.5)	16.4 (10.4)	8.9 (6.0)	t = 8.63 (df = 152.1)	<0.0001
Frequency (drinking occasions/week)	5.0 (1.2)	3.2 (1.3)	4.6 (1.4)	2.2 (1.0)	t = 20.801 (df = 157.46)	<0.0001
Binge occasions (no. per week)	1.5 (1.4)	1.2 (1.1)	1.4 (1.4)	1.0 (0.86)	t = 3.6129 (df = 153.12)	0.0004
Average eBAC ^d^ per week	0.033 (0.028)	0.019 (0.017)	0.031 (0.027)	0.018 (0.017)	t = 5.5123 (df = 153.36)	<0.0001
Peak eBAC ^e^ within past month	1.53 (1.09)	1.48 (0.78)	1.52 (1.04)	1.34 (0.87)	t = 2.0771 (df = 160.1)	0.004
Percent (%) over weekly recommended level	81 (68.1%)	13 (48.1%)	94 (64.4%)	575 (28.5%)	Z = 9.0714	<0.0001
Motivation to change drinking behavior (scale 0–10)	5.1 (2.7)	3.9 (3.0)	4.8 (2.7)	4.0 (2.6)	t = 3.5199 (df = 164.37)	0.0006

^a^: The data presented on continual and itinerant frequent-heavy drinkers are descriptive only and are not included in the statistical comparisons shown. ^b^: The term “moderate drinkers” is used here to describe the participants in seven of the eight hidden classes who were non-excessive or excessive drinkers who preferred to drink on one day a week. The statistical comparisons in this table are between two groups only—total frequent-heavy drinkers (*n* = 146) and participants with moderate drinking, a group that thus included some participants with excessive weekly drinking who were not members of the frequent-heavy group. ^c^: *p*-values are based on Welch two-sample *t*-test for age, AUDIT, quantity, frequency, binge occasions, average eBAC, and peak eBAC; a Z-score proportions test was used for gender and for percent over weekly recommended level. ^d^: Estimated average percentage blood alcohol count (BAC) per week. ^e^: Estimated average peak percentage BAC within the past month.

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
