# Peer review of "Smartphone Apps Targeting Hazardous Drinking Patterns among University Students Show Differential Subgroup Effects over 20 Weeks: Results from a Randomized, Controlled Trial"

_jcm, 2019, doi:10.3390/jcm8111807_

Round 1
Reviewer 1 Report
The revised manuscript by Berman et al. doesn’t seem to address my previous concern. I looked for a “response to reviewers,” but found none. If the authors address this concern in a response, I apologize for not finding it. At any rate, in figure 2 there are 722 subjects in the party planner group. In the three subsequent boxes it is stated that a total of 891 are “lost to follow-up”. This obviously cannot be correct. I suspect that this is just the number not responding on that particular week, but some subjects who don’t respond at the 7-week follow-up may respond in the subsequent 14 or 21 week follow-ups. If so, they should be referred to as “non-responders week 7”, etc. and not “lost to follow-up” that implies they were never seen again.
Author Response
Response to comments by Reviewer 1, revision 2:
[1.1] The revised manuscript by Berman et al. doesn’t seem to address my previous concern. I looked for a “response to reviewers,” but found none. If the authors address this concerns in a response, I apologize for not finding it.
Response: The authors had attached a 30-page document providing detailed answers to the editors' and both reviewers' comments. This was attached to the Cover letter as per the managing editor’s instructions. For Reviewer 1's information, we now attach our responses to their comments for the prior revision 1, with our responses in italics (see attachment).
[1.2] At any rate, in figure 2 there are 722 subjects in the party planner group. In the three subsequent boxes it is stated that a total of 891 are “lost to follow-up”. This obviously cannot be correct. I suspect that this is just the number not responding on that particular week, but some subjects who don’t respond at the 7-week follow-up may respond in the subsequent 14 or 21 week follow-ups. If so, they should be referred to as “non-responders week 7”, and not “lost to follow-up” that implies they were never seen again.
Response: We appreciate the reviewer’s repetition of this comment, and agree. The wording has been changed from ”Lost to follow-up” to “Non-responders”.

Reviewer 2 Report
There is an interesting paper addressing hazardous drinking in students and its possible modulation with smartphone apps. The subject is novel and innovative, however the paper is difficult to read and understand, possibly because the design is complicated and the authors showed results that should be added to results previously published.
Different points are detailed below in order to improve the manuscript.
Abstract:
The first paragraph needs to be rewritten, explaining clearly what the authors have done. Previously, a phrase introducing heavy alcohol drinking in university students and its modulation with app should improve the abstract.
Introduction:
Page 2. Lines 48-50. The hazardous alcohol consume it is also very frequent in the early and late adolescence. This type of consume is also related to a higher risk to onset a mental health disorders which could be comorbid with a substance use disorder. Something similar should be stated in the introduction section.
Page 3. Paragraph 2. Line 124-130. The authors should clarify the objectives of the study. Objective 1 has to be more explained, since it is new data. Moreover, it should be placed firstly, since objective 2 make reference to unpublished results form a large study. The design is so complicated and must be easily explained to readers.
Method:
Page 5. It could be considered a reward / incentive the fact to be included in a lottery for to participate in the study? If this is so, it should be considered in the discussion of the results, as a limitation or make an explanation.
Results:
Where authors have been considered prior differences in frequency of drinking for PArtyPlanner group?
What means inconclusive results regarding relative risk overt 7 weeks for PratyPlanner group?
Discussion:
The authors have not discussed the results obtained. For example, have not explained the relationship founded between level of motivation and alcohol consume? The same results were observed with other type of treatments (not TIC ) were done? … Or in the case of the profile obtained with the hidden Markov Model Analysis (males,..). This profile has been related with the later development of alcohol use disorder? Or whit more severe negative consequences?... The differences obtained between excessive and non-excessive drinkers in the different apps used should also be related with literature about the subject, for example the efficacy of the skill-training programs in alcohol treatments.
Author Response
Responses to comments by Reviewer 2:
[2.1] There is an interesting paper addressing hazardous drinking in students and its possible modulation with smartphone apps. The subject is novel and innovative, however the paper is difficult to read and understand, possibly because the design is complicated and the authors showed results that should be added to results previously published.
Response: We appreciate the reviewer’s detailed suggestions on how to improve the manuscript and have detailed our responses below.
[2.2] Abstract. The first paragraph needs to be rewritten, explaining clearly what the authors have done. Previously, a phrase introducing heavy alcohol drinking in university students and its modulation with apps should improve the abstract.
Response: The abstract has largely been re-written (see tracked changes in the manuscript) and a phrase introducing hazardous as well as heavy drinking among university students has been added at the start of the abstract.
[2.3] Introduction. Page 2, Lines 48-50. The hazardous alcohol consume it is also very frequent in the early and late adolescence. This type of consume is also related to a higher risk to onset a mental health disorder which could be comorbid with a substance use disorder. Something similar should be stated in the introduction section.
Response: The first sentence in the introduction has been lengthened as follows, and we have added a phrase indicating that adolescence also is a time for high alcohol consumption. We did not include early adolescence, however, since we have not targeted individuals under 18 in our research. See tracked changes in manuscript and below.
Evidence has accumulated showing that overconsumption of alcohol, particularly heavy drinking over time, is associated with a high burden of disease with a range of negative personal, social and economic effects, including higher risk of mental health conditions and high comorbidity of severe mental health disorders [1,2].
This type of behavior is most frequent among in adolescence and among young adults 18-29 [6] who are in a stage of “emerging adulthood” characterized by challenges of identity exploration, and management of emotional and circumstantial instability [7].
[2.4] Paragraph 2. Line 124-130. The authors should clarify the objectives of the study. Objective 1 has to be more explained, since it is new data. Moreover, it should be placed firstly, since objective 2 make reference to unpublished results form a large study. The design is so complicated and must be easily explained to readers.
Response: We thank the reviewer for identifying an area where our explanation of the design was still confusing. We have placed the aims (Objectives) first in the final paragraph of the Introduction, as suggested. We have also added text clarifying that only the third, HMM analysis includes study participants that participated in the already published study [27].
[2.5] Method: Page 5. It could be considered a reward / incentive the fact to be included in a lottery for the participate in the study? If this is so, it should be considered in the discussion of the results, as a limitation or make an explanation.
Response: The reviewer seems to be suggesting that the use of a lottery is questionable as it may have biased the sample. The chance of winning the lottery was infinitesmal in this case, and we have previously used this method in several studies with university students.
However, to clarify we added the following sentence to the Experimental section under 2.1, Procedure.
“A lottery is one form of compensation commonly used as an incentive to increase response rates. In this study, all participants had an equal opportunity to win the lottery if they completed all follow-ups.”
[2.6] Results. Where authors have been considered prior differences in frequency of drinking for PartyPlanner group?
Response: Indeed, there were slight a priori differences in drinking frequency for the PartyPlanner group in relation to controls, where PartyPlanner participants drank less frequently than controls. However, what is important regarding the 7-week follow-up is the changes in frequency between groups, where no difference was found over time between PartyPlanner and controls in the mixed model analysis. The a priori difference has no importance in this context, and we therefore do not comment on it. See pedagogical diagram (see pdf attachment) for the frequency variables by group over the 7-week period.
[2.7] What means inconclusive results regarding relative risk over 7 weeks for PratyPlanner group?
Response: To clarify, the wording has been changed from “inconclusive” to the following: “In comparisons between the PartyPlanner group and controls, there were no significant differences; see Table 2.”
The word “inconclusive” was originally used in order to conform to the possibility that Bayesian analyses could have shown differences that were not detected using the p-value based statistical analyses we used.
[2.8.A] Discussion: Have not explained the relationship founded between level of motivation and alcohol consume?
Response: We had already included text concerning the relationship between motivation and alcohol consumed, under Conclusion, originally lines 68-73, lines 86-89 as well as 92-94; see below for quotes. In order to clarify that our statements are part of the discussion section, we have move the Conclusion section to the last paragraph in the article, beginning “In summary…”.
”Second, the importance of individual motivation to reduce drinking for positive effects suggests that apps should include a measure of motivation to change and offer feedback indicating that higher motivation is associated with actual positive change. Initial information on app use could include a warning that short-term use may be associated with negative or null results, and that longer-term use is recommended. Non-motivated users might benefit from motivational app interventions as a precursor to use of apps with the type of content we evaluated.”
Also, we discuss the role of motivation in future research, in lines 86-89 as well as 92-94.
” For university students, a prior study has demonstrated positive effects of combining motivational counseling with feedback to reduce heavy drinking among students [43], suggesting a possible catalytic role for human guidance in combination with digital interventions for university students.”
“In summary, future research should begin to disentangle differential effects of eBAC and skills-based apps, more extended digital programs and guidance versus no guidance, in relation to baseline characteristics including severity of alcohol use, drinking patterns, and motivation.”
[2.8.B] The same results were observed with other type of treatments (not TIC) were done? …
Response: To clarify, we added the following text on line 69: “in combination with eBAC apps”, see tracked changes.
[2.8.C] Or in the case of the profile obtained with the hidden Markov Model Anaysis (males,..). This profile has been related with the later development of alcohol use disorder?
Response: The following text in italics has been added, on lines 83-84, see tracked changes.
”Future studies for both student and adult populations with hazardous and excessive drinking should apply innovative design to offer differential interventions for varying drinking patterns, paying particular attention to the needs of frequent-heavy excessive drinkers, whose alcohol consumption is likely to be associated with the development of Alcohol Use Disorder and related comorbid physical and mental conditions.”
[2.8.D] Or whit more severe negative consequences? …
Response: See response to [2.8C], in italics, “related comorbid physical and mental conditions.”
[2.8.E] The differences obtained between excessive and non-excessive drinkers in the different apps used should also be related with literature about the subject, for example the efficacy of the skill-training programs in alcohol treatments.
Response: We have inserted a text on this on lines 78-81, “This is in concordance with current research on digital interventions for problematic alcohol use showing that more intense programs, including modules on refusal skills and management of negative emotions, tend to have slightly better effects on problematic use [15,16].”

Round 2
Reviewer 2 Report
I'm agree with all the changes maded.
This manuscript is a resubmission of an earlier submission. The following is a list of the peer review reports and author responses from that submission.
Round 1
Reviewer 1 Report
The manuscript by Berman et al. reports on the results of a clinical trial looking at the effectiveness of smartphone apps that look to reduce problem drinking. The current report follows up on a previous publication that reported finding from individuals in the current trial that were identified as excessive drinkers at the 7-week follow-up. I am not sure why the authors chose to publish the work in this manner, and I don’t think it is the ideal approach, but it does appear that there is enough new here to support a separate publication. Overall the manuscript is well written and the results presented clearly. Given the large number of these apps that are now on the market, it is important that there be scientific investigations of the apps effectiveness. This research will provide an important baseline for future research in this area.
My only real concern with the manuscript is the large drop-out rate. In this regard Figure 2 is also confusing. If looks like more participants dropped out of the Party Planner group than were enrolled. I’m obviously missing something here. Even for the other groups there is a large drop-out rate. There might be a lot of self-selection for more motivated participants.
Reviewer 2 Report
THis is a potentially interesting study in an area that needs further research. However, I found the Introduction hard to follow, and the delineation between past use of the present data and the current study unclear. I also have queries regarding the analytic approach. I detail main concerns below.
Introduction
Page 1, line 44: Unclear what is meant by 'personalized (normative)' - this seems oxymoronic to me.
Page 2, lines 53: 'evidence ... is lacking'. It is unclear whether the authors mean that there have been few (if any) attempts to empirically evaluate these apps, or whether the existing evidence does not support their use. I suspect the latter, though this should be made clearer to the reader.
Several lines later, the authors mention that the research to date makes their question 'open'. It is unclear how this would be the case as the phrase 'open question' to me would imply either: (i) the prior results are very inconsistent, to the point where it is hard to ascertain whether the intervention is efficacious; or (ii) there is not enough research to date to make a strong conclusion either way.
The argument from line 58 to 62 seems contradictory.
Some discussion of actual effect sizes from this prior work - especially towards the end of the Introduction - would be worthwhile.
I am unclear about the contribution of this study beyond the prior publication arising from these data. It seems that the prior publication was focused on heavy drinkers - wouldn't this be the group that is logically of most interest? If focusing on other participants in this study who didn't meet the excessive drinker threshold, is that still meaningful? At the very least, I think a stronger justification is necessary here.
Method
7. Line 28, p.6: is the intervention app or web based? not clear.
8. How accurate is the BAC rating from your app?
9. Is it meaningful to cluster by universities if there are only six groups at this level? What are the implications for variance and for statistical power?
10. For the HMM, have you taken into consideration time trends due to intervention AND also due to day of the week?
11. Some analysis or justification of the MAR assumption for missing data is necessary.